# Redefining the specificity of phosphoinositide-binding by human PH domain-containing proteins

Nilmani Singh [1,6], Adriana Reyes-Ordoñez[1,6], Michael A. Compagnone[1], Jesus F. Moreno[1], Benjamin J. Leslie[2], Taekjip Ha [2,3,4,5] & Jie Chen [1✉]

Pleckstrin homology (PH) domains are presumed to bind phosphoinositides (PIPs), but specific interaction with and regulation by PIPs for most PH domain-containing proteins are unclear. Here we employ a single-molecule pulldown assay to study interactions of lipid vesicles with full-length proteins in mammalian whole cell lysates. Of 67 human PH domain-containing proteins initially examined, 36 (54%) are found to have affinity for PIPs with various specificity, the majority of which have not been reported before. Further investigation of ARHGEF3 reveals distinct structural requirements for its binding to PI(4,5)P$_2$ and PI(3,5)P$_2$, and functional relevance of its PI(4,5)P$_2$ binding. We generate a recursive-learning algorithm based on the assay results to analyze the sequences of 242 human PH domains, predicting that 49% of them bind PIPs. Twenty predicted binders and 11 predicted non-binders are assayed, yielding results highly consistent with the prediction. Taken together, our findings reveal unexpected lipid-binding specificity of PH domain-containing proteins.

[1] Department of Cell & Developmental Biology, University of Illinois at Urbana-Champaign, Urbana, IL, USA. [2] Department of Biophysics and Biophysical Chemistry, Johns Hopkins University School of Medicine, Baltimore, MD, USA. [3] Department of Biophysics, Johns Hopkins University, Baltimore, MD, USA. [4] Department of Biomedical Engineering, Johns Hopkins University, Baltimore, MD, USA. [5] Howard Hughes Medical Institute, Baltimore, MD, USA. [6]These authors contributed equally: Nilmani Singh, Adriana Reyes-Ordoñez. ✉email: jiechen@illinois.edu

Specific phospholipid–protein interactions are critical to the regulation of many signal transduction processes and cellular functions. These interactions typically involve lipid-binding domains recognized by specific lipid species and/or physical properties of the membrane such as charge or curvature[1–4]. The largest family of putative lipid-binding domains (LBDs) is the pleckstrin homology (PH) domain, with over 250 members encoded by the human genome (Pfam database[5]). Originally defined by its presence in the protein Pleckstrin[6,7], this domain of 100–120 amino acids has an invariable structure of seven-stranded β-sandwich lined by a C-terminal α-helix[8]. PH domains are found in many different types of proteins that are involved in regulating diverse signaling pathways and functions. Some of the earliest characterized PH domains, including those in Pleckstrin, RasGAP, and GRK2, were found to have affinity for PI(4,5)$P_2$[9]. The PH domains in PLCδ and AKT1 bind with high selectivity to PI(4,5)$P_2$ and PI(3,4,5)$P_3$ (PIP$_3$), respectively[10,11]. Indeed, the interaction with PIP$_3$ is so specific for those two PH domains that they have been commonly used as sensors to detect PI(4,5)$P_2$ and PIP$_3$ in cells[12,13].

Despite those early examples of PH–PIP interactions, to date only a modest number of PH domain-containing proteins have been demonstrated to bind PIPs with specificity. A comprehensive analysis of *S. cerevisiae* PH domains revealed that only one of them had specific affinity for a particular PIP and the rest of them displayed little affinity or selectivity for PIPs[14]. Another study examined a large number of mouse PH domains and found 20% of them to translocate to the plasma membrane (PM) in response to PIP$_3$ production in cells, but most of those PIP$_3$-responsive PH domains did not show specific binding to PIP$_3$ in lipid binding assays[15]. Thus, it has been proposed that the specific PIP recognition by PLCδ-PH and AKT1-PH may be the exception rather than the rule for PH domains and that only a small percentage of all PH domains bind PIPs with high affinity and specificity[4]. Protein partners have been identified for PH domains, which may cooperate with lipid–PH interaction or operate independently to regulate or mediate PH domain functions[1,16].

It is important to note that our knowledge to date of the lipid binding properties of PH domain-containing proteins is largely derived from studies of isolated PH domains rather than full-length proteins, at least partly owing to the hurdle of purifying full-length proteins for lipid binding assays. Intra-molecular interaction is a universal mechanism for determining protein structure and activity, and this crucial determinant would be eliminated when a PH domain is taken out of the context of the protein.

To circumvent the laborious process of purifying proteins and to preserve post-translational modifications of the proteins, we have developed a total internal reflection fluorescence (TIRF) microscopy-based single-molecule pulldown (SiMPull) assay to study protein–lipid interactions with whole cell lysates and lipid vesicles[17], herein referred to as lipid-SiMPull or SiMPull. The sensitivity and specificity of this assay have been demonstrated through highly selective pulldown of various LBDs, as well as full-length AKT1, from cell lysates by vesicles containing PIPs known to interact with those proteins[17]. We have now applied the lipid-SiMPull assay to investigate nearly 100 human full-length PH domain-containing proteins for their ability to bind vesicles of various compositions, including all seven types of PIPs. Furthermore, we have used the assay results to generate and validate a prediction algorithm through probabilistic modeling of amino acids sequence of PH domains, which reveals PH domain sequence determinants for PIP binding and predicts PIP binding for the entire human family of proteins with PH domains. Our findings reveal unexpected lipid binding specificity and suggest that PIP recognition by PH domain-containing proteins is more prevalent than previously believed.

## Results

**Lipid-SiMPull assay for human PH domain-containing proteins**. In order to investigate binding of PIPs by PH domain-containing proteins using the lipid-SiMPull assay[17], we created cDNA constructs and transfected them in human embryonic kidney (HEK) 293 cells to express 67 full-length human PH domain-containing proteins with EGFP fused at their C-termini. Expression of each fusion protein in HEK293 cells was confirmed by western blotting (see examples in Supplementary Fig. 1a). Concentrations of EGFP-fusion proteins in the lysates were determined by using a standard curve for fluorescence emission (Supplementary Fig. 1b). Small unilamellar vesicles were made with 60 mol % phosphatidylcholine (PC), 15 mol% phosphatidylserine (PS), 20 mol % cholesterol, and 5 mol% of one of seven PIPs: PI(3)P, PI(4)P, PI(5)P, PI(3,4)$P_2$, PI(4,5)$P_2$, PI(3,5)$P_2$, and PI(3,4,5)$P_3$. Vesicles containing 5% phosphatidic acid (PA), a different type of signaling lipid than PIPs, were also included in our assays. As a negative control, vesicles were made to include an additional 5 mol% PC in place of PIP (referred to as PC or control vesicles hereafter). All vesicles also had 0.01 mol % biotinylated phosphatidylethanol (PE) to facilitate immobilization on imaging slides.

SiMPull assays were performed with the vesicles immobilized on slides and lysates containing 5 nM EGFP-fusion protein flowed into the slide chamber followed by TIRF imaging (Fig. 1a). Each of the 67 proteins was assayed against each of the nine types of vesicles. The vesicles also contained a lipophilic red dye, DiD, to allow visualization of the immobilized vesicles on slides and confirmation of consistent vesicle density across assays. Pulldown of EGFP-tagged proteins was quantified by counting fluorescent spots on the surface. Each vesicle-protein pair was assayed with at least three independent lysates. Based on characterization of many positive and negative controls with the SiMPull assay, we arrived at 100 EGFP spots per area of 1600 μm$^2$ (after background subtraction) as the threshold for binding and the data were interpreted as a binary outcome––binding or no binding. As shown in Supplementary Data 1, the vast majority of the assay results from three independent experiments were remarkably consistent for the 67 proteins.

**Over half of the proteins examined bound PIP with specificity**. Previously we had shown that the SiMPull assay detected LBD–lipid interactions that were reported to have $K_d$'s in the high nanomolar/sub-micromolar range[17]. To estimate the threshold of detection under our assay conditions, we performed SiMPull with the PX domain of p40Phox, the WT, R60A, and K92A mutant of which were reported to bind PI(3)P with $K_d$ of 5 μM, 17.5 μM, and >50 μM, respectively[18]. As shown in Supplementary Fig. 2, robust binding to PI(3)P, but not to PI(4,5)$P_2$, was detected for WT p40Phox-PX. Binding to the R60A mutant was also observed, at just above the cut-off, whereas the K92A mutant did not bind the lipid. Therefore, our assay under current conditions has a detection threshold of affinity in the 10–20 μM range.

Assay results for the 67 proteins are summarized in Table 1 and Table 2, and the raw data can be found in Supplementary Data 1. For 10 proteins reported to bind PI(4,5)$P_2$ and/or PIP$_3$ but not found to bind lipids in our initial assay, we also performed the assay with vesicles containing 20% of the PIP. As shown in Table 1, 36 of the 67 proteins (54%) were found to bind PIPs with various specificity and none bound PA; four proteins displayed promiscuous binding to lipids including PA and PC, which could reflect misfolding of those proteins. The positive binding data are also illustrated in a binary table in Fig. 1b. Representative SiMPull assay images and quantification are shown in Fig. 1c for seven proteins, each displaying specificity for a different PIP.

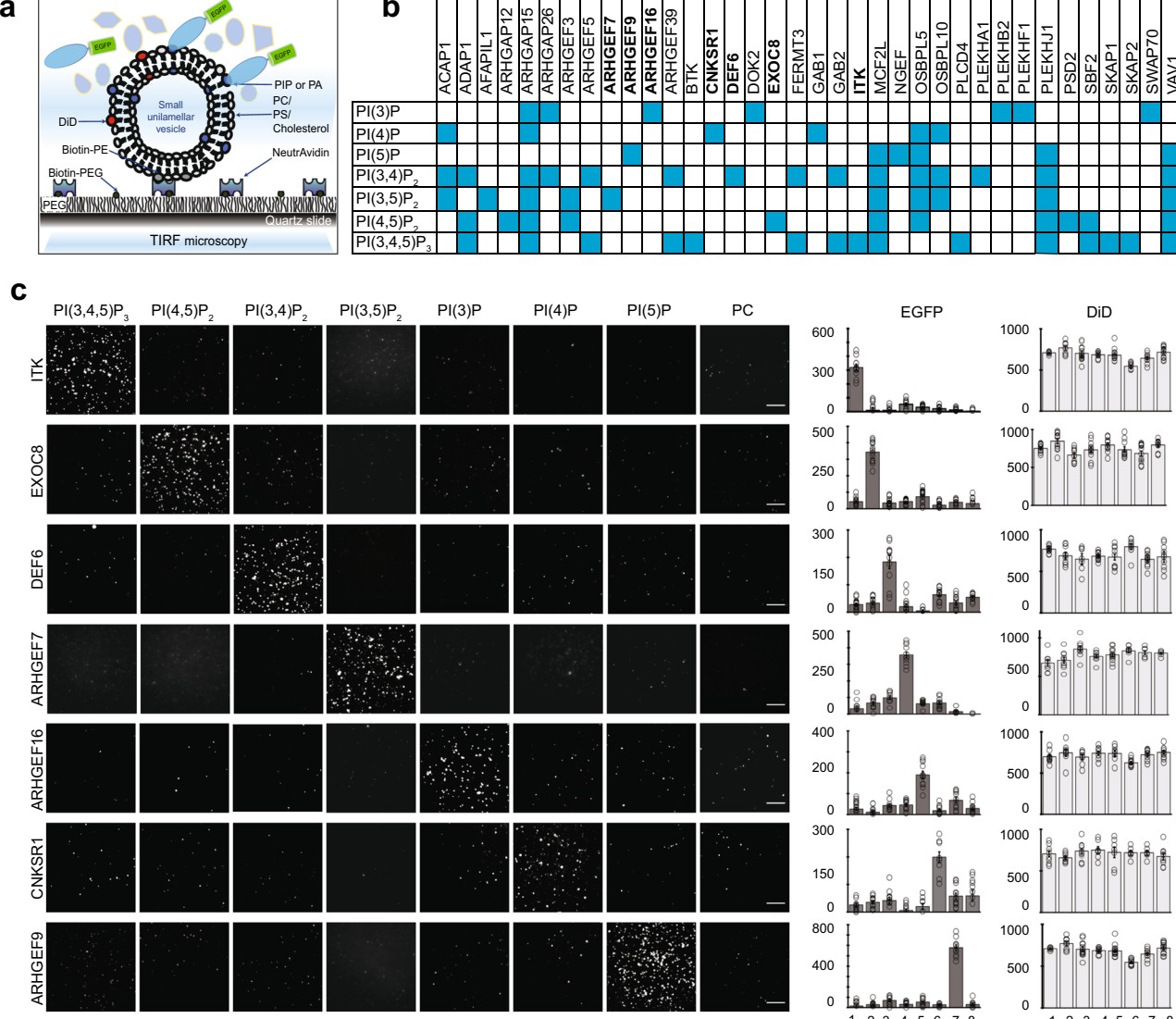

**Fig. 1 Specific PIP binding by full-length human PH domain proteins. a** Schematic representation of the lipid-SiMPull assay to detect lipid–protein interactions. Adapted with permission from Arauz et al.[17] Copyright 2016, American Chemical Society. **b** A binary representation of results for the 36 proteins found to bind PIPs in SiMPull. The vesicles are identified by the unique PIP, but all contained PC, PS, cholesterol, DiD, and biotin-PE. Blue: binding; white: no binding. SiMPull data for the proteins in bold are shown in (**c**). **c** Representative TIRF images of seven EGFP-fusions pulled down by vesicles containing 5% PIP, with PC as a negative control. Scale bars: 5 μm. Numbers of fluorescent molecules per image area are shown in the graphs on the right. Data are presented as mean ± SEM. $N \geq 10$ images for EGFP; $N \geq 6$ images for DiD. Source data are provided as a Source Data file, which lists the exact $N$ number for each data point. 1–8: PIP$_3$, PI(4,5)P$_2$, PI(3,4)P$_2$, PI(3,5)P$_2$, PI3P, PI4P, PI5P, PC. Each assay was repeated with lysates from at least three independent transfections, and the data can be found in Supplementary Data 1. The results for 67 proteins are summarized in Table 1 & 2.

**Phosphatidylserine is not responsible for the PH domain protein–PIP interactions.** A majority of the 36 proteins that bound PIPs in our assays are either not reported to bind lipids or reported to bind lipids with different specificity. It is believed that different species of phospholipids may cooperate in the membrane to determine specific interaction with proteins[2,3]. For instance, PS has been found to contribute to enhanced affinity and/or specificity for PIPs by yeast PH domains[19]. Since the vesicles in our assays contained 15 mol% PS (based on mammalian cell membrane composition), we wondered whether the newly discovered PIP–PH protein interactions were dependent on PS. To address this possibility, we re-assayed 23 of the PIP-binding proteins with vesicles containing only PC and a specific PIP, omitting PS and cholesterol. Remarkably, all 23 proteins bound their respective PIPs without PS in the vesicles

(Fig. 2a). Representative SiMPull images are presented in Fig. 2b. Therefore, the presence of PS does not appear to account for the PIP–PH protein interactions emerged from our studies.

**Distinct structural requirements for ARHGEF3 binding to PI(4,5)P$_2$ and PI(3,5)P$_2$.** To compare the behaviors of PH domains to their full-length protein counterparts, we performed SiMPull assays with EGFP fusions of 12 PH domains from 10 proteins found to bind PIP in our assays (two of them have two PH domains each). To our surprise, with the exception of PLEKHA1, the PH domains either did not bind PIP or bound different PIPs than their full-length counterparts (Supplementary Table 1). This discrepancy could be attributed to misfolding of the PH domains when expressed alone and/or interference by the EGFP tag, although it is also possible that the full-length proteins have

**Table 1 Lipid-SiMPull assay results for 40 PIP-binding PH domain-containing proteins.**

| Protein name | Entrez gene ID | PIPs bound |
|---|---|---|
| BTK | 695 | PI(3,4,5)P3 |
| ITK | 3702 | PI(3,4,5)P3 |
| SKAP1 | 8631 | PI(3,4,5)P3 |
| *PLCD4 | 84812 | PI(3,4,5)P3 |
| *SKAP2 | 8935 | PI(3,4,5)P3 |
| EXOC8 | 149371 | PI(4,5)P2 |
| PSD2 | 84249 | PI(4,5)P2 |
| *ARHGAP12 | 94134 | PI(4,5)P2 |
| DEF6 | 50619 | PI(3,4)P2 |
| PLEKHA1 | 59338 | PI(3,4)P2 |
| AFAP1L1 | 134265 | PI(3,5)P2 |
| ARHGEF7 | 8874 | PI(3,5)P2 |
| ARHGEF16 | 27237 | PI(3)P |
| DOK2 | 9046 | PI(3)P |
| PLEKHB2 | 55041 | PI(3)P |
| PLEKHF1 | 79156 | PI(3)P |
| SWAP70 | 23075 | PI(3)P |
| CNKSR1 | 10256 | PI(4)P |
| GAB1 | 2549 | PI(4)P |
| ARHGEF9 | 23229 | PI(5)P |
| NGEF | 25791 | PI(5)P |
| *SBF2 | 81846 | PI(3,4,5)P3, PI(4,5)P2 |
| ARHGEF39 | 84904 | PI(3,4,5)P3, PI(3,4)P2 |
| FERMT3 | 83706 | PI(3,4,5)P3, PI(3,4)P2 |
| GAB2 | 9846 | PI(3,4,5)P3, PI(3,4)P2 |
| ARHGEF5 | 7984 | PI(4,5)P2, PI(3,4)P2 |
| ARHGEF3 | 50650 | PI(4,5)P2, PI(3,5)P2 |
| ARHGAP26 | 23092 | PI(3,4)P2, PI(3)P |
| ACAP1 | 9744 | PI(3,4)P2, PI(3,5)P2, PI(4)P |
| OSBPL10 | 114884 | PI(3,4)P2, PI(3,5)P2, PI(4)P |
| ADAP1 | 11033 | PI(3,4,5)P3, PI(4,5)P2, PI(3,4)P2 |
| MCF2L | 23263 | PI(3,4,5)P3, PI(4,5)P2, PI(3,4)P2, PI(3,5)P2, PI(5)P |
| OSBPL5 | 114879 | PI(4,5)P2, PI(3,4)P2, PI(3,5)P2, PI(4)P, PI(5)P |
| PLEKHJ1 | 55111 | PI(3,4,5)P3, PI(4,5)P2, PI(3,4)P2, PI(3,5)P2, PI(5)P |
| VAV1 | 7409 | PI(3,4,5)P3, PI(4,5)P2, PI(3,4)P2, PI(3,5)P2, PI(5)P |
| ARHGAP15 | 9938 | PIP3, PI(4,5)P2, PI(3,4)P2, PI(3,5)P2, PI(3)P, PI(4)P |
| PLEKHO1 | 51177 | PI(3,4,5)P3, PI(4,5)P2, PI(3,4)P2, PI(3)P, PI(5)P, PA |
| CERT1 | 10087 | all vesicles tested |
| NET1 | 10276 | all vesicles tested |
| SPATA13 | 221178 | all vesicles tested |

The proteins are sorted in the order of the number of specific PIPs they bind. *Binding was observed with 20% PIP and not 5% PIP; only PI(3,4,5)P3 and PI(4,5)P2 were assayed at 20%. See Supplementary Table 1 for data.

**Table 2 PH domain-containing proteins found to not bind any PIP in the initial lipid-SiMPull assays.**

| Protein name | Entrez gene ID |
|---|---|
| AFAP1 | 60312 |
| APPL1 | 26060 |
| APPL2 | 55198 |
| ARHGAP25 | 9938 |
| BMX | 660 |
| DOK1 | 1796 |
| DOK3 | 79930 |
| DOK4 | 55715 |
| FERMT2 | 10979 |
| FGD5 | 152273 |
| GRB7 | 2886 |
| GRB14 | 2888 |
| GRK2 | 156 |
| KIF1B | 23095 |
| OSBPL8 | 114882 |
| PHETA2 | 150368 |
| PLCG2 | 5336 |
| PLEK | 5341 |
| PLEKHF2 | 79666 |
| PLEKHN1 | 84069 |
| PLEKHO2 | 80301 |
| PRKD3 | 23683 |
| PSD4 | 23550 |
| RASA1 | 5921 |
| RASGRF1 | 5923 |
| SOS2 | 6655 |
| STAP1 | 26228 |

(Fig. 3b). We constructed EGFP-PH fusions with two slightly different sequence definitions for the ARHGEF3 PH domain and each tagged by EGFP at either terminus, and subjected them to SiMPull assays against PI(4,5)P2. The PH domain defined by the reported crystal structure[22] bound the lipid, whereas the sequence defined by Uniprot (slightly larger) did not result in binding (Supplementary Fig. 3). We then further assayed the PH constructs that bound PI(4,5)P2 against all the other PIPs. As shown in Fig. 3c, the PH domain with EGFP fused at the N-terminus bound a variety of PIPs including PIP3, PI(3,4)P2, PI(4,5)P2, PI4P and PI5P, but with EGFP at its C-terminus the PH domain bound only PI(4,5)P2; neither constructs bound PI(3,5)P2. We went on to perform additional domain analysis of ARHGEF3 for binding to PI(4,5)P2 and PI(3,5)P2. As shown in Fig. 3d, deletion of the PH domain (ΔPH) abolished binding to both lipids. Interestingly, deletion of the N-terminal 125 amino acids (ΔN) also abolished binding to both lipids, whereas deletion of both N- and C-termini (DH-PH) restored PI(4,5)P2 binding but not PI(3,5)P2 binding. On the other hand, a fragment composed of the N-terminal domain and the PH domain bound both lipids. Expression of the various ARHGEF3 fragments was confirmed by western blotting shown in Supplementary Fig. 4a. Taken together, our data suggest that the PH domain is necessary and sufficient for ARHGEF3 binding to PI(4,5)P2, whereas PI(3,5)P2 binding requires the N-terminal domain in addition to the PH domain. The C-terminus appears to suppress ARHGEF3 binding to both lipids, and this suppression may be relieved by the N-terminal domain.

**Bacterially expressed ARHGEF3 binds PI(4,5)P2 but not PI(3,5)P2.** Due to overexpression of the EGFP-fusion proteins it is unlikely that any endogenous proteins in the lysate would reach stoichiometric levels to contribute directly to the observed

unique structural features that determine their PIP binding that are not recapitulated by the isolated PH domains. Regardless of the explanation, our observations suggest that caution should be taken to interpret results of protein–lipid interactions based on the commonly employed EGFP-tagged PH domains.

To gain deeper mechanistic insight, we selected a novel PIP binder on our list to perform detailed analysis: ARHGEF3 (also named XPLN for exchange factor found in platelets, leukemic, and neuronal tissues), a GEF for RhoA and RhoB[20]. ARHGEF3 (Fig. 3a) belongs to the Dbl family of 70 human RhoGEFs, the vast majority of which contain a Dbl homology (DH) domain and a PH domain in tandem[21]. There had been no report of specific lipid binding by ARHGEF3. In our assay, the full-length ARHGEF3 protein bound specifically to PI(4,5)P2 and PI(3,5)P2

**a**

| Protein | Lipid |
|---------|-------|
| ACAP1 | PI(4)P |
| ARHGAP15 | PI(3,4)P$_2$ |
| ARHGAP26 | PI(3,4)P$_2$ |
| ARHGEF3 | PI(4,5)P$_2$ |
| ARHGEF5 | PI(3,4)P$_2$ |
| ARHGEF9 | PI(5)P |
| ARHGEF16 | PI(3)P |
| ARHGEF39 | PI(3,4)P$_2$ |
| CNKSR1 | PI(4)P |
| DEF6 | PI(3,4)P$_2$ |
| DOK2 | PI(3)P |
| EXOC8 | PI(4,5)P$_2$ |
| GAB1 | PI(4)P |
| MCF2L | PI(4,5)P$_2$ |
| NGEF | PI(5)P |
| OSBPL5 | PI(4,5)P$_2$ |
| OSBPL10 | PI(3,4)P$_2$ |
| PLEKHB2 | PI(3)P |
| PLEKHJ1 | PI(3,4,5)P$_3$ |
| PLEKHO1 | PI(3,4,5)P$_3$ |
| PSD2 | PI(4,5)P$_2$ |
| SWAP70 | PI(3)P |
| VAV1 | PI(4,5)P$_2$ |

**b**

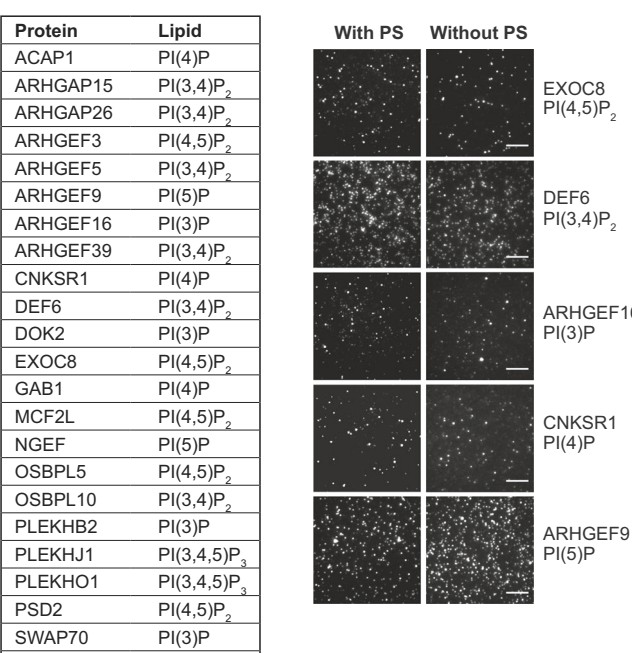

**Fig. 2 The presence of PS in lipid vesicles does not affect PIP binding by proteins. a** SiMPull was performed for 23 proteins against lipid vesicles containing the indicated PIPs, with or without PS and cholesterol. Binding was observed for all with both types of vesicles. For each condition at least two independent experiments were performed with similar results. Data were analyzed as described for Table 1 and Supplementary Data 1. **b** Representative TIRF images of EGFP pulldown from a are shown for five proteins that are also shown in Fig. 1c, each by vesicles with or without PS. Scale bars: 5 μm.

recombinant protein binding to lipid vesicles. Nevertheless, we wished to test this directly with ARHGEF3 and a few other proteins by expressing the EGFP-fusion proteins in *E. coli*. To date, we tested 11 full-length proteins from our PIP-binding list, but all of them presented serious challenges for purification. Solubility of the proteins was poor even when expressed at low temperature (18 °C). Various fusion tags were tested and the SUMO tag yielded the best solubility, consistent with a previous report[23]. However, the proteins tended to crash out of solution upon purification. Hence, we devised a SiMPull assay using bacterial lysates expressing the proteins of interest. Bacteria cells expressing the recombinant proteins were lysed, and the lysates were cleared by ultracentrifugation, measured for EGFP concentration, and subjected to SiMPull against various PIPs at 5 nM EGFP-protein. The rationale is, it is highly unlikely that *E coli* and human cells would share any protein cofactors that may facilitate PIP binding. Of the 11 proteins examined, seven proteins were found to bind nonspecifically to lipid vesicles including the negative control PC, probably reflecting misfolding of the proteins. We pursued the remaining four proteins with SiMPull assays, and found that two of them (ARGEF16 and ITK) bound PIP with specificity that matched their mammalian counterparts (Supplementary Fig. 5). A third protein, DEF6, bound both PI(3,4)P$_2$ and PIP$_3$ although in mammalian cells it bound only PI(3,4)P$_2$. We still consider this result a reasonable validation because these two PIPs tend to have overlapping affinity for the same proteins. Inerestingly, bacterially expressed ARHGEF3 bound PI(4,5)P$_2$ with specificity but not to PI(3,5)P$_2$. Taking into consideration the domain analyses described above, we propose that the full-length ARHGEF3 may require another factor to

relieve an autoinhibition on PI(3,5)P$_2$ binding. An alternative and equally likely explanation would involve a posttranslational modification of ARHGEF3 necessary for PI(3,5)P$_2$ binding and not PI(4,5)P$_2$ binding.

**PI(4,5)P$_2$ binding is necessary for ARHGEF3 membrane targeting, activation of RhoA, and induction of stress fiber formation.** In reported PH domain–PIP headgroup interactions, negatively charged amino acids in the β1-β2 and β3-β4 loops play critical roles[1]. Applying the Patch Finder Plus 2.3 software[24] to the reported ARHGEF3-PH crystal structure[22], we identified a positively charged patch containing K342, R345, K348, R376 and H427 to be potentially involved in the interaction between ARHGEF3 and PIP (Fig. 4a). These amino acids were mutated to alanine or acidic residues in full-length ARHGEF3 individually and in combination, and the mutant proteins (Supplementary Fig. 4a, b) were subjected to SiMPull assays with PI(4,5)P$_2$ vesicles. Mutation of any one of the positively charged residues did not result in drastic reduction of PI(4,5)P$_2$ binding, but three double-mutants (K342E/K348E, R345D/R376D, R345D/H427D) and a triple-mutant (R345A/R376A/H427D) each abolished lipid binding by ARHGEF3 (Fig. 4b). Next, we investigated PI(3,5)P$_2$ binding by the ARHGEF3 mutants. As shown in Fig. 4b, the triple mutant and R345D/H427D abolished PI(3,5)P$_2$ binding. Interestingly, the K342E/K348E mutant retained PI(3,5)P$_2$ binding activity. These observations suggest that the PH domain of ARHGEF3 may interact with the two PIPs via structurally distinct mechanisms.

Because PI(4,5)P$_2$ is mostly found in the plasma membrane (PM), we wondered if ARHGEF3 binding to PI(4,5)P$_2$ played a role in the protein's membrane recruitment. Recombinant EGFP-ARHGEF3 was found throughout the cell including the nucleus, and the nuclear localization is consistent with the presence of a nuclear localization sequence in ARHGEF3. We observed prominent presence of EGFP-ARHGEF3 in the PM of NIH3T3 cells, and the PI(4,5)P$_2$ binding-deficient mutant of ARHGEF3, K342E/K348E displayed drastically reduced localization to the PM (Fig. 5a). This observation is consistent with ARHGEF3 targeting to the PM via its interaction with PI(4,5)P$_2$.

Next, we asked whether PI(4,5)P$_2$ binding is relevant to ARHGEF3's function in a cell. ARHGEF3 has two unrelated activities: as a GEF for RhoA/B[20] and as a GEF-independent inhibitor of mTORC2 phosphorylation of AKT[25]. We examined these activities of ARHGEF3 in HEK293 cells. As shown in Fig. 5b, expression of wild-type ARHGEF3 suppressed AKT phosphorylation as expected, and expression of each lipid binding-deficient mutant of ARHGEF3 (K342E/K348E, R345D/R376D, R345D/H427D, or R345A/R376A/H427D) was equally effective in inhibiting AKT, suggesting that PI(4,5)P$_2$ binding is not involved in ARHGEF3's function as an inhibitor of mTORC2 (nor is PI(3,5)P$_2$ binding). This is consistent with the previous observation that the N-terminal 125-amino acid fragment, devoid of the PH domain, is sufficient to inhibit mTORC2[25]. On the other hand, unlike the wild-type protein, the K342E/K348E mutant ARHGEF3 was ineffective in activating cellular RhoA activity upon overexpression, as measured by pulldown of active RhoA using an effector protein (Fig. 5c). Hence, lipid binding is apparently necessary for ARHGEF3's GEF activity toward RhoA in the cell.

We also performed in vitro GEF assays with ARHGEF3 and ARHGEF3-K342E/K348E proteins purified from bacteria, and found that the two proteins displayed near identical activities towards RhoA (Fig. 5d). We were not able to assess the effect of lipids on the GEF activity because we found addition of lipid vesicles to be incompatible with the GEF assay. Nevertheless, our observation confirmed that the K342E/K348E mutant retained

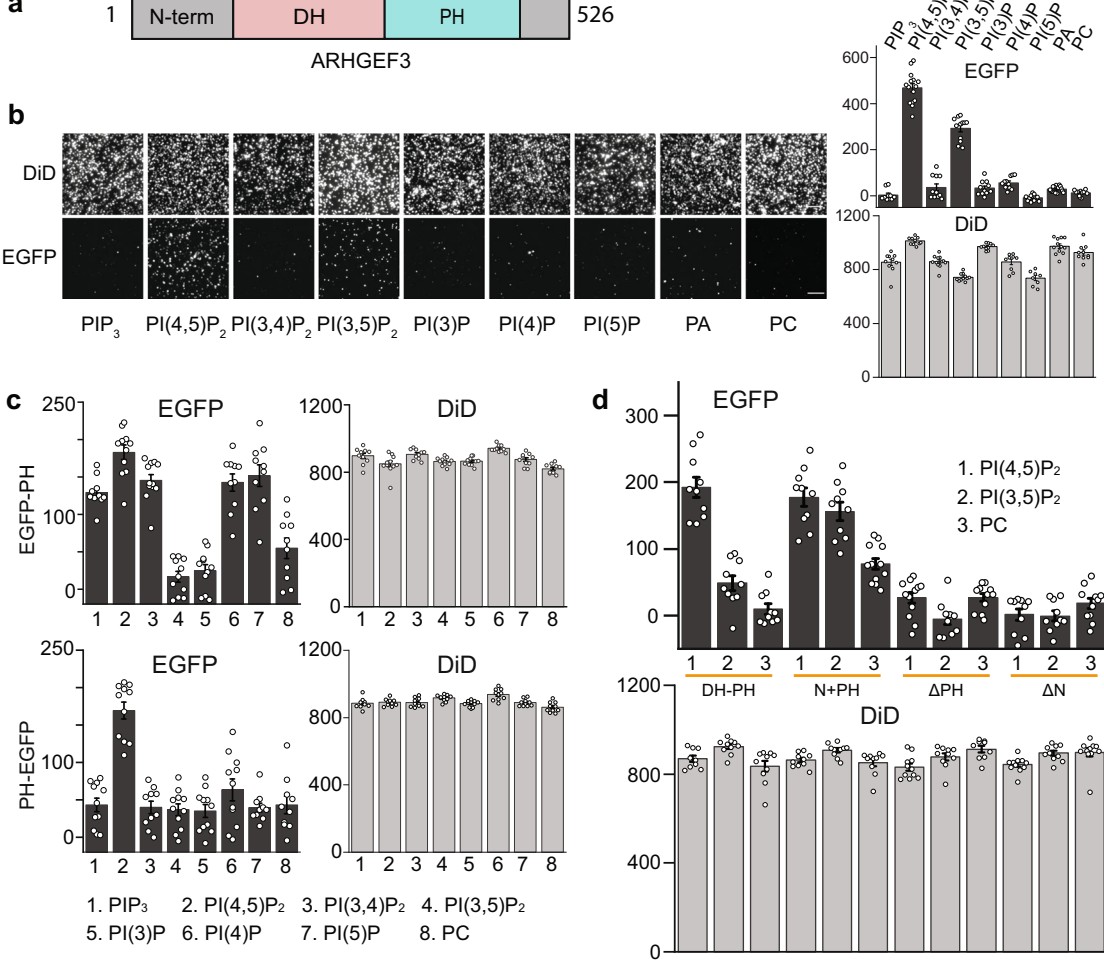

**Fig. 3 Domain analysis of ARHGEF3 binding to PI(4,5)P$_2$ and PI(3,5)P$_2$. a** Schematic diagram of ARHGEF3 domain structure. **b** HEK293 cells were transiently transfected with EGFP-ARHGEF3, and the lysates were subjected to SiMPull assays against various types of lipid vesicles as indicated. Representative TIRF images are shown. Scale bars: 5 μm. Numbers of fluorescent molecules per image area are shown in the graphs on the right. Each assay was performed with at least three independent transfections with similar results. **c** Similar to (**b**), ARHGEF3 PH domain tagged by EGFP at N- or C-terminus was assayed for binding to seven types of PIP vesicles and PC control vesicles. **d** Similar to (**b**), various EGFP-tagged ARHGEF3 fragments were assayed for binding to vesicles containing 5% PI(4,5)P$_2$. DH-PH: aa125–455; N + PH: aa1-125 + aa320-455; ΔPH: aa304-466 deleted; ΔN: aa118–526. All data in graphs are presented as mean ± SEM. $N \geq 10$ images for EGFP; $N \geq 6$ images for DiD. Source data are provided as a Source Data file, which lists the exact $N$ number for each data point. Each assay was performed with at least three independent transfections with similar results.

proper folding and enzymatic activity. The simplest explanation for all our results combined is that PI(4,5)P$_2$ binding and/or PM targeting is necessary to achieve maximum GEF activity for ARHGEF3 in the cell.

Consistent with its role as a GEF for RhoA, ARHGEF3 has been reported to regulate actin cytoskeleton reorganization and assemble stress fibers in the cell[20]. To probe a potential role of ARHGEF3-PI(4,5)P$_2$ interaction in this process, we expressed EGFP-ARHGEF3 in NIH3T3 cells and visualized actin cytoskeleton with phalloidin staining. As shown in Fig. 5e, expression of wild-type ARHGEF3, but not the K342E/K348E mutant, led to robust stress fiber formation, suggesting that PI(4,5)P$_2$ binding may be critical for ARHGEF3 regulation of actin cytoskeleton reassembly, a RhoA-dependent process. Taken together, our observations uncover a mechanism of ARHGEF3 regulation and provide functional validation of a protein-lipid interaction identified in our lipid-SiMPull assays.

**Recursive functional classification identifies potential amino acid determinants of PIP binding distributed throughout the**

**entire PH domain.** Although our survey of 67 proteins covered only 1/4 of the PH domain-containing protein family in the human genome, we reasoned that our results could have predictive power for the rest of the family. Given the high likelihood that the PH domains mediate PIP binding of the full-length proteins, we compared the sequences of the PH domains of the 67 proteins we studied plus AKT1-PH as a positive control, excluding the four proteins that bound lipids non-specifically. We also excluded two proteins (ADAP1 and AFAP1L1) that each contained two PH domains and bound PIP as full-length proteins, because we did not know which PH domain contributed to the binding. This yielded a collection of 67 PH domain sequences. Based on reported structural studies of the PH domain-PIP headgroup interaction, the loop between the first two β strands (β1-β2) with a basic sequence motif of "KX$_n$(K/R)XR" plays the most prominent role[3]. However, we found the vast majority of the 67 PH domains to contain this motif, with no distinction between those that bound PIPs and those that did not (Supplementary Fig. 6). Therefore, we next considered the possibility that sequence features throughout the entire PH domain may be determinants of PIP binding.

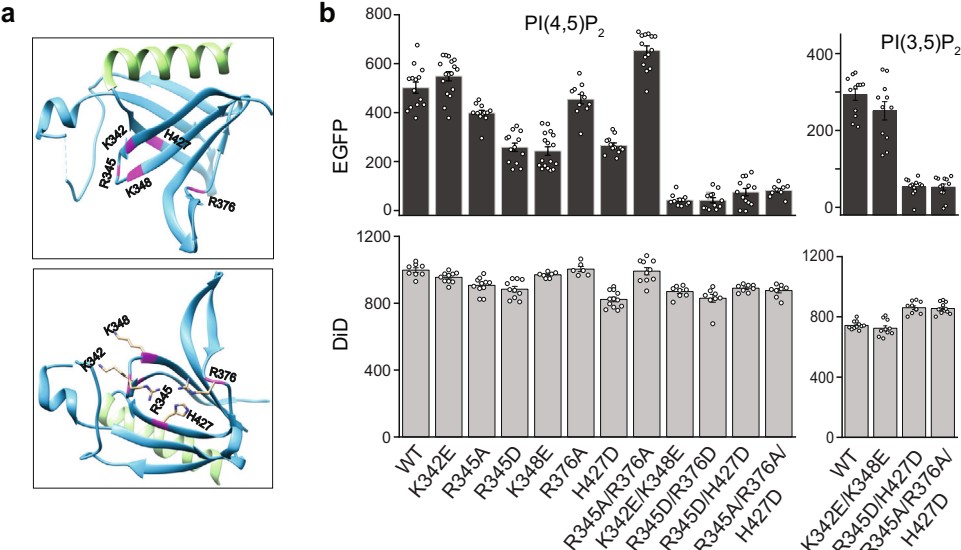

**Fig. 4 Mutations in the PH domain differentially abolish ARHGEF3 binding of PI(4,5)P$_2$ and PI(3,5)P$_2$. a** Ribbon representation of the structure of ARHGEF3 PH domain (PDB ID: 2Z0Q;[22] image created with Pymol v2.5), with residues predicted to interact with PIP highlighted in magenta and the sidechains of those residues shown in the lower panel. **b** HEK293 cells were transiently transfected with EGFP-ARHGEF3 containing various mutations at the residues highlighted in (**a**), and the lysates were subjected to SiMPull assays against vesicles containing PI(4,5)P$_2$ or PI(3,5)P$_2$. Numbers of fluorescent molecules per image area are shown in the graphs. Data are presented as mean ± SEM. $N \geq 10$ images for EGFP; $N \geq 6$ images for DiD. Source data are provided as a Source Data file, which lists the exact $N$ number for each data point. Each assay was performed with at least three independent transfections with similar results.

To identify sequence signatures of PIP binding, we employed an unbiased probabilistic sequence comparison strategy––recursive functional classification (RFC)[15]. Sequence alignment was performed for 242 PH domains from 220 human proteins, including the 67 PH domains described above. The alignment was achieved with the PROMALS3D (PROfile Multiple Alignment with predicted Local Structures and 3D constraints) tool, which makes use of available 3D structural information[26]. PH domains are known to have low sequence conservation but they all share a conserved structure. The incorporation of structural information is therefore particularly important for the accurate alignment of PH domain sequences. The sequence alignments for the 67 PH domains investigated in SiMPull and the other 175 PH domains are shown in Supplementary Data 2 and Supplementary Data 3, respectively.

The 67 assayed PH domains (35 bound and 32 did not bind PIP) were used to create an RFC matrix, which could be used to calculate the predicted PIP binding score ($S_{RFC}$) for any PH domain, as previously described[15]. The resulting scores separated the 35 PH domain bound PIPs from the 32 that did not bind (Fig. 6a). Importantly, the separation improved progressively as the region included in the calculation of $S_{RFC}$ was expanded from β1–β2 loop alone, to β1–β2, β3–β4, and β6–β7 loops, to the entire PH domain. The relative contributions of various regions of the PH domain to PIP-binding varied drastically among those found to bind PIPs (Fig. 6b), suggesting that the structural basis of PH–PIP interactions may be diverse within this family of proteins. A heat map (Fig. 6c) was created to represent the RFC-matrix for the 67 assayed PH domains aligned to the sequence of AKT1-PH based on its crystal structure (PDB ID: 1UNP[27]). Amino acids with high impact on lipid binding, either positively or negatively, were distributed throughout the PH domain. The overall contribution of each amino acid position to PIP binding is shown in Fig. 6d, with 12 positions of the highest contribution marked. While some of those residues are located in the classically defined head group binding pocket, many are not.

Other than those 12, additional positions spanning the entire PH domain are likely to also play important roles in PIP binding (Fig. 6d). It should be noted that this is a composite view, and that an individual PH domain would use only a subset of those positions for interaction with PIP (see Fig. 6b).

**The RFC algorithm predicts 50% of human PH domain-containing proteins to bind PIP.** Next, we applied the RFC algorithm obtained above to analyze the human PH domains not investigated by SiMPull. While in principle any protein with a positive $S_{RFC}$ score can be considered to potentially bind PIP, we set out to establish a more stringent cut-off score for positive correlation with PIP binding by scrambling the sequences of all 242 PH domains and subjecting them to scoring by the same algorithm (Supplementary Fig. 7). Using mean +3x SD, we arrived at 2.59 as the threshold $S_{RFC}$ score for a PH domain to be considered likely to bind PIP. Of the 175 PH domains not tested by SiMPull, 86 (49%) scored above the threshold (Fig. 7a and Supplementary Data 3). Taking into consideration our assay data, the overall projection is that 50% of the PH domains in the human proteome bind PIPs.

To test the accuracy of the RFC prediction algorithm, we performed SiMPull assays with 20 predicted binders that had not been reported to bind any lipids and 11 predicted non-binders. The availability of cDNA clones and confirmation of EGFP-fusion full-length protein expression led to this random selection of 31 proteins (Fig. 7b,c and Supplementary Data 4). For the predicted binders, because binding to any PIP by a protein would validate the prediction for that protein, we assayed those 20 proteins against one of the three PIPs frequently found to bind PH domain-containing proteins in our assays: PIP$_3$, PI(3,4)P$_2$, and PI(4,5)P$_2$. If binding was confirmed for one PIP, that protein was not assayed against the other PIPs, but all the proteins were assayed against the negative control vesicles (PC) to rule out non-specific binding. As presented in Fig. 7b,c and Supplementary Data 4, 19 out of the 20 proteins were found to bind at least one

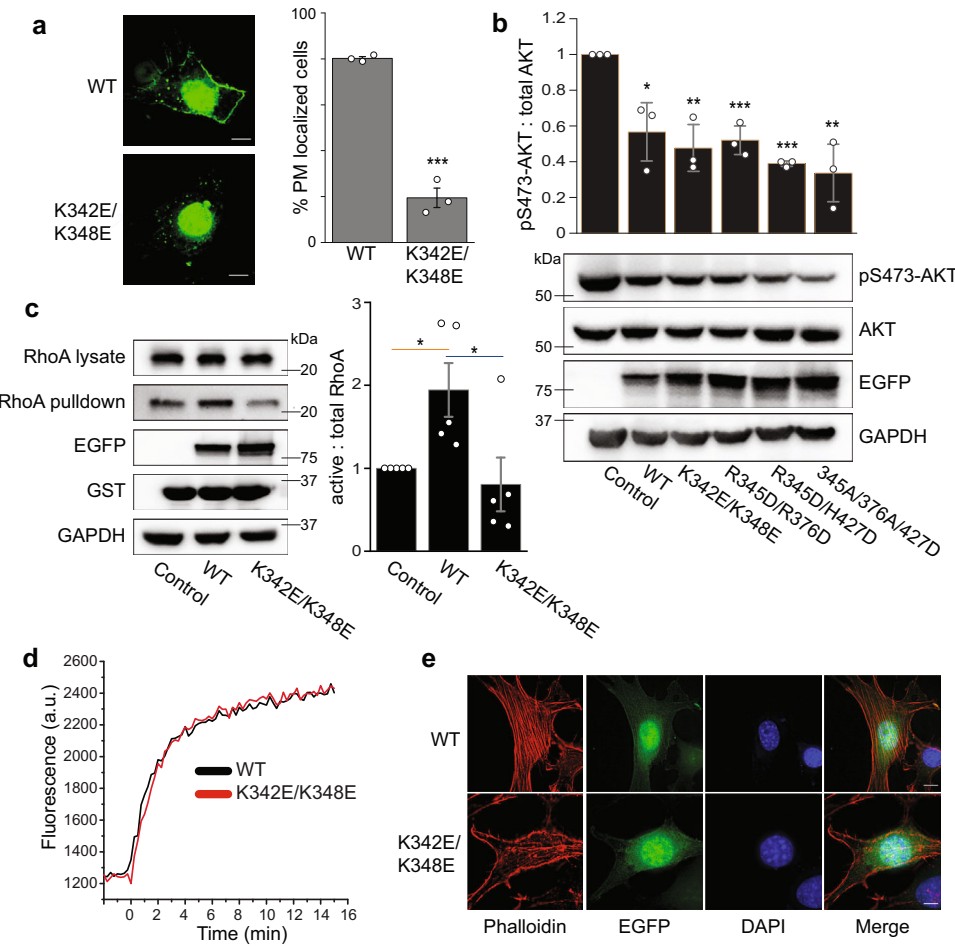

**Fig. 5 PI(4,5)P$_2$ binding is necessary for ARHGEF3 membrane targeting, cellular activity and function. a** NIH3T3 cells were transfected for 24 h with EGFP-tagged wild-type or lipid binding-deficient mutant K342E/K348E of ARHGEF3. The transfected cells were then treated with 0.03% digitonin for 2.5 min to remove cytosolic proteins before fixation. Percentage of transfected cells with PM localization was quantified. Data are presented as mean ± SEM ($n = 3$). Scale bars: 5 μm. **b** HEK293 cells were transfected with wild-type or lipid binding-deficient mutants of ARHGEF3, serum-starved overnight, and then stimulated with 10% FBS for 30 min followed by cell lysis and western blotting. Western blot signals were quantified by densitometry to generate ratio of pAKT versus total AKT. Data are presented as mean ± SEM ($n = 3$ independent experiments). **c** HEK293 cells were transfected for 24 h with empty vector (control), wildtype, or K342E/K348E mutant of ARHGEF3. Cell lysates were subjected to active RhoA pulldown assay using GST-rhotekin beads, and analyzed by western blotting. Western blot signals were quantified by densitometry to generate the relative ratios of RhoA pulled down (active) and RhoA in lysates (total). Data are presented as mean ± SEM ($n = 5$ independent experiments). **d** Purified wild-type and K342E/K348E mutant ARHGEF3 were subjected to in vitro RhoA guanine nucleotide exchange assay. Three independent experiments were performed with similar outcome, and representative results are shown. **e** NIH3T3 cells were transfected for 24 h with wild-type or K342E/K348E mutant of ARHGEF3, followed by fixation and phalloidin/DAPI staining. Three independent experiments were performed with similar outcome, and representative images are shown. Scale bars: 5 μm. Two-tailed Student's *t* test was performed to compare WT and mutant in (**a**), and each sample to the control in (**b**, **c**). *$P < 0.05$; **$P < 0.01$; ***$P < 0.001$. *P* values: **a** 0.0001. **b** 0.0164, 0.0039, 0.0008, 0.0000, 0.0039. **c** 0.020, 0.038. Source data for (**a–d**) are provided as a Source Data file.

of the initial three PIPs. PSD3, which did not bind any PIP in the initial assays, was further assayed against the other four PIPs and not found to bind any of them. For the predicted non-binders, each protein was assayed against all seven PIPs plus the negative control. Eight of those proteins did not bind any PIP whereas 3 bound at least one PIP. In retrospect, one of the three proteins that bound PIP, CYTH3, has an unusual PH domain structure that contains nine β-sheets instead of the typical 7, which might have led to askew RFC analysis. In summary, 19 out of 20 predicted binders were validated and 8 out of 11 predicted non-binders were validated. The fact that experimentally we found PIP binders among predicted non-binders suggests that PIP binding by PH domain family proteins may be even more prevalent than the 50% projected by our current data and modeling. Based on these results, a confusion matrix was constructed (Fig. 7d), and various metrics were calculated to evaluate our model: Accuracy

= 0.87, Recall = 0.864, Precision = 0.95 and F-1 score = 0.90. These results validate the predictive power of our recursive learning approach and, furthermore, are consistent with the notion that the PH domain defines the observed PIP binding by full-length proteins.

## Discussion

Our analysis of a random collection of 67 human PH domain-containing proteins using the lipid-SiMPull assay with a detection threshold of $K_d$ in the 10–20 μM range reveals that 54% of them bind PIPs and that many of the specific interactions have not been reported before. The vast majority of PH domain–PIP interactions reported have affinities well within the detection threshold of our assay[28,29]. Hence, it is unlikely that our assay has missed many true interactions although it is possible that some physiologically relevant interactions, such

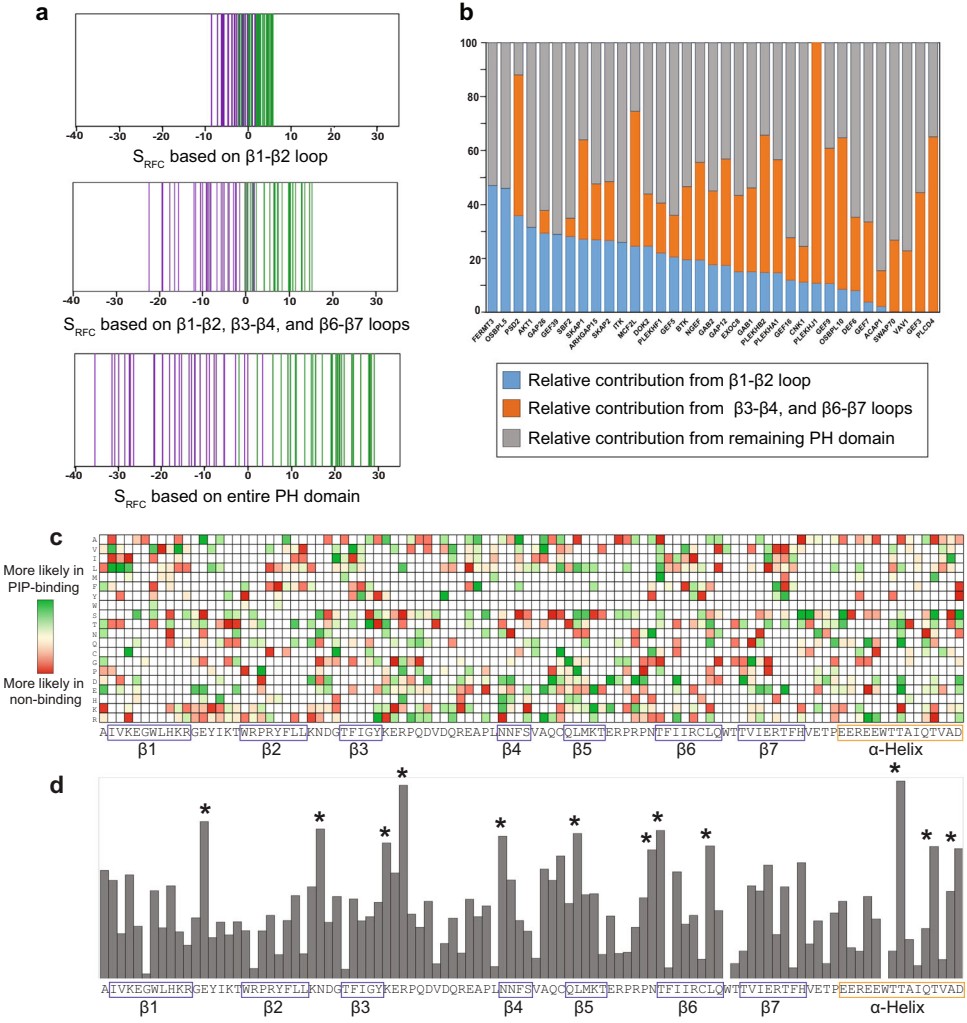

**Fig. 6 Recursive-learning algorithm distinguishes PIP-binding when taking the entire PH domain into consideration. a** $S_{RFC}$ scores were calculated for the 67 PH domains in our assays, with amino acids from β1–β2 variable loop, β1–β2, β3–β4, β6–β7 variable loops, or the entire PH domain. Each vertical line represents one PH domain. Green: bound PIP in SiMPull assays; purple: did not bind PIP in SiMPull assays. **b** Relative contribution by β1–β2 loop, β3–β4, and β6–β7 loops, or the rest of the PH domain is plotted based on $S_{RFC}$ for each protein that bound PIP in SiMPull assays. **c** A heat map showing the contribution of 20 amino acids at each position for the 67 PH domains. The amino acid positions are aligned to AKT1-PH domain based on its crystal structure (PDB ID: 1UNP[27]). **d** Contribution from different positions of PH domain towards PIP-binding was plotted from RFC-matrix. Twelve positions with the highest scores are marked by stars and aligned to the sequence in (**c**). Source data are provided as a Source Data file.

as those involving additional domains and proteins[30], are not detected. As an example of functional validation of a previously-not-reported interaction that emerged from our assays, we have demonstrated that PI(4,5)P$_2$ binding is necessary for the GEF activity of ARHGEF3 in the cell. Taking advantage of the assay data, we have created a recursive-learning algorithm, which has predicted 49% of the remaining human PH domains in the family to bind PIPs with some specificity. The predictive power of this algorithm is validated by our assay results of 20 predicted binders (none of which had been reported to bind PIPs previously) and 11 predicted non-binders, which yields a precision of 95% and recall of 86%. Altogether we present here PIP binding assays with 98 full-length PH domain-containing proteins, and conclude that at least 50% of all human PH domain-containing proteins likely bind PIPs with specificity, in contrast to the previous estimate that only 10% of all PH domains bind PIPs with high affinity and specificity[4].

The SiMPull assay offers the unique advantage of using full-length proteins expressed in the native environment (i.e.,

mammalian cells) without purification. These proteins would be likely to maintain their native structures and proper post-translational modifications. Because the EGFP-fusion proteins were overexpressed in cell lysates and any potential endogenous protein partners may not reach the necessary stoichiometric level, the lipid–protein interactions observed in lipid-SiMPull assays are most likely intrinsic properties of the PH domain-containing proteins. Our successful experimental validation of PIP binding predicted by the recursive learning algorithm based on prob-abilistic comparison of PH domain sequences also supports the notion that the PH domain defines PIP binding by the full-length proteins. Nevertheless, possible involvement of endogenous mediators of PIP binding cannot be ruled out. Our attempt to bacterially express these PH domain-containing full-length pro-teins was largely unsuccessful most likely due to misfolding of the recombinant proteins. Of the four proteins potentially well-folded when expressed in bacteria, three recapitulated PIP binding in mammalian cell lysates, whereas bacterially expressed ARHGEF3 bound only one of the two PIPs interacted with the protein expressed in mammalian cells. This is consistent with the idea

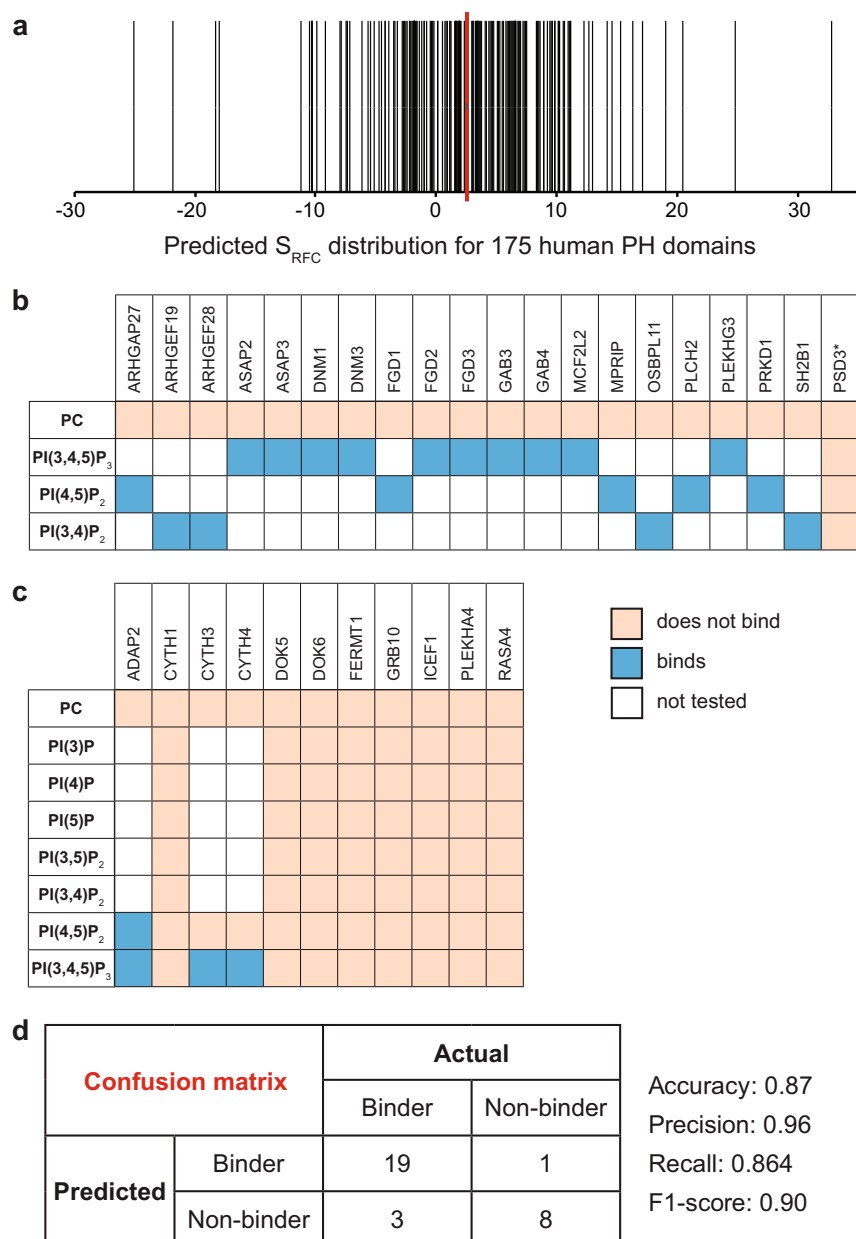

**Fig. 7 RFC prediction of PIP binding by human PH domain-containing proteins and validation of the predictions. a** Prediction of PIP-binding for 175 human PH domains using the RFC-matrix. Each vertical black line represents one PH domain. The red line indicates the cutoff value (2.59) for PIP-binding. **b** A binary representation of results of SiMPull assays for 20 predicted PIP binders against various vesicles. *PSD3 was assayed against all 7 PIPs and did not bind any. Blue: binding; peach: no binding; white: not tested. **c** A binary representation of results of SiMPull assays for 11 predicted PIP non-binders against various vesicles. Blue: binding; peach: no binding; white: not tested. **d** A confusion matrix constructed with results in (**b**, **c**), from which various parameters for accuracy metrics are shown. Source data for a are provided as a Source Data file.

that the protein expressed in bacteria may lack either a necessary post-translational modification or a co-factor. Previous large-scale lipid binding studies relied on isolated PH domains (not full-length proteins) that were either bacterially expressed and purified[14,19], or expressed in mammalian cells but applied to the lipid strips assay that is known to be highly unreliable[31], leading to some researchers' conclusion that better assays would be necessary[15]. A mass spectrometry study of proteins pulled down by PIPs immobilized on agarose beads identified 405 PIP-binding proteins in the human proteome, but only 33 PH domain-containing proteins were found among them[32]; while powerful, the lack of membrane environment for the PIPs in this method

may present a limitation. Our assay uses small unilamellar vesicles that mimic the composition of cell membranes, which are commonly employed to study lipid–protein interactions. However, we do not know whether they have the heterogeneity of native membranes with nanodomains that could influence the interactions. Future efforts will be needed to address this issue.

The vast majority of the 70 members of the Dbl family of RhoGEFs contain a PH domain immediately following the catalytic DH domain[21]. Although those PH domains had been speculated to bind phospholipids and subsequently contribute to membrane localization and/or allosteric regulation of the Rho-GEFs, very few of them have been reported to have marked

affinity for specific PIPs. Thus, the idea of phospholipids regulating RhoGEFs through their PH domains remains controversial[33]. Among the proteins we studied with the SiMPull assay, 14 are RhoGEFs belonging to the Dbl family. PIP binding was observed for nine of them (ARHGEF3, ARHGEF5, ARHGEF7, ARHGEF9, ARHGEF16, ARHGEF39, MCF2L, NGEF, VAV1), all of which were either never reported to bind lipids or reported to bind lipids with different specificity than what we observed. For instance, ARHGEF3 has not been reported to bind phospholipids with any specificity. We have found that full-length ARHGEF3 binds PI(4,5)P$_2$ and PI(3,5)P$_2$, and that the binding is dependent on the PH domain. Interestingly, a mutant (K342E/K348E) ARHGEF3 has lost PI(4,5)P$_2$ binding while retaining its binding of PI(3,5)P$_2$, thus offering an ideal tool to establish the importance of PI(4,5)P$_2$ binding for the GEF activity of ARHGEF3. Future investigation is warranted to probe a role of PI(3,5)P$_2$ in ARHGEF3 function. Follow-up studies on the 8 remaining RhoGEFs and investigation of lipid binding by other Dbl family members will also likely be illuminating.

Our RFC analysis suggests that for some proteins PIP binding may involve regions of the PH domain outside of the classic PIP headgroup-interacting site. Similar suggestions have been made by others for PIP$_3$ regulation[15] and organelle PIP binding[19] by PH domains. Interestingly, two most recent molecular dynamics simulation studies of the PH domain of GRP1 have revealed that there may be multiple PIP$_3$ binding sites––canonical and non-canonical––on this domain[34,35]. Furthermore, our results of domain analysis of ARHGEF3 suggest that PIP binding likely extends beyond the PH domain, and that PI(4,5)P$_2$ and PI(3,5)P$_2$ have different structural requirements for binding to ARHGEF3. Future biochemical and structural studies to probe these different modes of protein–lipid interactions are warranted.

## Methods

**cDNA cloning and mutagenesis.** Plasmids harboring cDNAs for human PH domain-containing proteins from human cDNA library hORFeome V5.1 were cloned as a pool into the pDest-eEGFP-N1 vector using Gateway Cloning, followed by identification of individual clones. Additional cDNA clones were obtained from DNASU, and cloned individually into the pcDNA3-EGFP vector using Gibson assembly. The PH domains (except ARHGEF3) were cloned into the pEGFP-C1 vector using Gibson assembly. ARHGEF3 (XPLN) expression plasmids for full-length and truncations had either been previously reported[25] or created in this study using pEGFP-C1 or pEGFP-N1 vector. The amino acid sequence boundaries of the various ARHGEF3 truncations are described in the legend of Supplementary Fig. 4. p40PhoxPX-EGFP was obtained from Addgene (#19010)[36]. All point mutants were created using QuikChange Lightning Site-Directed Mutagenesis Kit following the manufacturer's protocol. All cDNAs in the final plasmids were sequence-confirmed in their entirety.

**Cell culture.** HEK293 cells were maintained in high-glucose DMEM with 10% FBS, 2 mM L-glutamine, and penicillin/streptomycin at 37 °C in 5% CO$_2$. NIH3T3 cells were maintained in high-glucose DMEM with 10% newborn calf serum and 4 mM L-glutamine, and penicillin/streptomycin at 37 °C in 5% CO$_2$. For SiMPull experiments, HEK293 cells were transfected for 24 hours in six-well plates using PolyFect® (3 µL/µg DNA) following manufacturer's recommendations. For immunofluorescence experiments, cells were plated in 12-well plate on poly-L-lysine coated coverslips and transfected with Lipofectamine™ 3000 following manufacturer's recommendations. For active RhoA pulldown assays, HEK293 cells were transfected in 10-cm plates with Lipofectamine™ 3000.

**Cell lysis and western blotting.** For SiMPull assays, the cells were collected after 24-hr transfection in detergent-free buffer (40 mM HEPES, pH8.0, 150 mM NaCl, 10 mM β-glycerophosphate, 10 mM sodium pyrophosphate, 2 mM EDTA, 1x Sigma protease inhibitor cocktail). The cells were lysed by sonication for 3 seconds on ice followed by ultracentrifugation at 90,000 × g in a TLA100.3 rotor for 1 h at 4 °C. For SiMPull, EGFP concentration was measured using a standard emission (ex488/em520) curve of pure recombinant EGFP (see Supplementary Fig. 1B), and each cell lysate was diluted in vesicle buffer (10 mM Tris·HCl, pH 8.0, 150 mM NaCl) to yield 5 nM EGFP. For western blotting, cells were lysed in 1x SDS sample buffer or as described above and mixed at 1:1 with 2x SDS sample buffer, both containing β-mercaptoethanol at a final concentration of 5%, and heated for 5 min at 95 °C. Proteins were resolved by SDS-PAGE and transferred onto PVDF

membrane. The membrane was incubated with primary and secondary antibodies following manufacturers' recommendations. HRP-conjugated secondary antibody was reacted with West Pico PLUS Chemiluminescent Substrate, and the signal was detected using an iBright CL 1000 imaging system (Thermo Fisher Scientific).

**Protein expression in bacteria.** The E. coli strain BL21 was used for all bacteria protein expression in this study. GST-ARHGEF3 and GST-ARHGEF3-K342E/K348E in pGEX-4T1 were expressed by IPTG induction at 18 °C overnight and purified using glutathione Sepharose (GE Healthcare) following the manufacturer's recommendations. SUMO-EGFP-various cDNAs in pET-28a-SUMO were expressed by IPTG induction at 18 °C for 30 min. The cell pellets were resuspended in PBS containing 1 mg/mL lysozyme and frozen at −80 °C. Upon thawing the bacteria were lysed spontaneously, and the lysates were cleared by ultra-centrifugation as described earlier. EGFP concentrations were measured, and the lysates were diluted to 5 nM EGFP for SiMPull analysis.

**Lipid vesicle preparation.** All lipids were mixed in chloroform (0.166 µmol total) and dried under nitrogen flow. The dried mixture was re-suspended in 100 µL vesicle buffer (10 mM Tris·HCl, pH 8.0, 150 mM NaCl) to a final concentration of 1.66 mM. After 30 min incubation at room temperature, vesicles were formed by water bath sonication (Laboratory Supplies, Hicksville, NY, model G112SPIT, 600 v, 80 kc, and 0.5 A) in 4 cycles of 4 min each. Small unilamellar vesicles were collected as the supernatant after ultracentrifugation at 194,398 × g in a TLA100.3 rotor for 1 hr at 25 °C.

**Lipid-SiMPull assay.** Quartz slides were prepared as described in Jain et al. and Arauz et al.[17,37] as follows: the slides were thoroughly cleaned and passivated with PEG doped with 0.1–0.2% biotin-PEG. Neutravidin (200 µg/mL) was incubated in chambers for 10 min, followed by addition of biotinylated lipid-vesicles. Freshly prepared whole-cell lysate (80 µL) was added by flowing into each slide chamber, replacing the vesicle solution. An inverted total internal reflection fluorescence (TIRF) microscope with Olympus 100x, NA1.4 lens and EMCCD camera (Andor iXon Ultra 897) was used to acquire single-molecule data at 10 frames/second. Diode pumped solid state lasers were used to excite EGFP at 488 nm (Coherent) and DiD at 638 nm (Cobalt). All SiMPull experiments were performed at room temperature. TIRF images were acquired using the smFRET (2014) package and processed in IDL (6.2SE) to generate image files, and the number of fluorescence spots were identified with point-spread function. MATLAB®2016A was used to extract data from data files generated in IDL (6.2SE) and import into Microsoft Excel (2016). The background number of EGFP spots was gathered prior to addition of lysate and subtracted from EGFP spots after lysate addition to generate data for each image. At least 10 SiMPull images (1600 µm² each) were analyzed to generate the average number of EGFP spots per imaging area for each assay. The smFRET package and IDL scripts used to process raw image files are available from Github: https://github.com/Ha-SingleMoleculeLab and https://doi.org/10.5281/zenodo.4925617.

**Fluorescence imaging.** HEK293 or NIH3T3 cells on poly-L-lysine-coated glass coverslips were fixed in 3.7% paraformaldehyde at room temperature for 15 min and permeabilized with 0.1% Triton X-100 for 5 min. For visualizing stress fibers, Rhodamine phalloidin was diluted 1:3000 and incubated with 3% BSA/PBS at 4 °C for 60 min, followed by incubation with DAPI (1:2500) for 20 min at room temperature. For digitonin permeabilization experiments, cells on coverslips were treated with 0.03% digitonin (20 mM HEPES, PH 7.5, 110 mM KOAc, 5 mM NaOAc, 2 mM Mg(OAc)$_2$, 1 mM EGTA) for 2.5 min, followed by fixation with 3.7% paraformaldehyde. A personal deconvolution microscope system (DeltaVision, Applied Precision) was used with a 100x or 60x NA 1.4 lens to capture fluorescence images. Deconvolution used an enhanced ratio iterative–constrained algorithm[38]. The acquired images were processed in Fiji (ImageJ2). For quantification of PM-localized cells, 60-100 transfected cells were counted in each experiment.

**GTPase RhoA activity assays.** To measure cellular RhoA activity, HEK293 cells in 10-cm plates were transfected for 24 h and then lysed in 50 mM Tris (pH 7.4), 10 mM MgCl$_2$, 500 mM NaCl, 1% (vol/vol) Triton X-100, 0.1%SDS, 0.5% sodium deoxycholate, and 1x Protease Inhibitors cocktail. The cell lysates were incubated for 1 hr at 4 °C with 60 µg of GST-rhotekin beads, followed by washing with 50 mM Tris (pH 7.4), 10 mM MgCl$_2$, 150 mM NaCl, 1% (vol/vol) Triton X-100, and 1x Protease Inhibitors cocktail, and analysis by western blotting with an anti-RhoA antibody. To measure nucleotide exchange activity of RhoA in vitro, the mant-GTP exchange assay was performed following manufacturer's recommendation in 20 µL reaction volume in a 384-well plate. In short, purified RhoA (1 µM) was added to exchange buffer (40 mM Tris-HCl, pH 7.5, 100 mM NaCl, 20 mM MgCl$_2$, 4 µM mant-GTP) and fluorescence (ex360/em440) was recorded for 5 min before adding purified GST-ARHGEF3 or GST-ARHGEF3-K342E/K348E.

**RFC matrix generation and PIP-binding prediction.** The list of human PH domain-containing proteins was obtained from the Pfam database and individual

PH domain sequences were acquired from UniProt (https://www.uniprot.org/). After initial alignment of all sequences using PROMALS3D with default parameters, 34 sequences were removed due to large insertions, and the remaining 242 PH domains were re-aligned. MATLAB®2016A was used to create two $297 \times 20$ (MSA positions x amino acids) probability matrices for PIP-binding ($P^B$) and non-binding proteins ($P^{NB}$), respectively. An amino acid was scored in the probability matrix if it was found in at least one PH domain in the group. Based on the assumption that each sequence position contributes independently to PIP binding, an RFC matrix was created with each element of the matrix calculated as $\log(P^B/P^{NB})$. In Fig. 5b, the value at each amino acid position is the average of sum-square of values in the corresponding position. This RFC-matrix was used to calculate the predicted PIP binding score ($S_{RFC}$) for any PH domain: the values in the RFC matrix for matching amino acids at all positions were summed. The source code for RFC matrix and PIP binding prediction is available on Github: https://doi.org/10.5281/zenodo.4927511.

**Reporting summary**. Further information on research design is available in the Nature Research Reporting Summary linked to this article.

## Data availability
All data generated or analyzed during this study are included in the manuscript, its supplementary information, and source data file, and from the corresponding author upon reasonable request. Source data underlying all figures, and the original western blotting images (iBright, Thermo Fisher Scientific) are provided with this manuscript. Source data are provided with this paper.

## Code availability
All codes created for or used in this study are available from Github and Zenodo. smFRET package used for SiMPull image acquisition: https://github.com/Ha-SingleMoleculeLab; IDL scripts used to process raw image files: https://doi.org/10.5281/zenodo.4925617; source code for RFC matrix and PIP binding prediction: https://doi.org/10.5281/zenodo.4927511.

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

## Acknowledgements
We thank Dr. Lisa Stubbs and Mr. Joe Troy for help with the initial identification of PH domain sequences, and Ms. Lucy Yao and Ms. Eleanor Marcet for assistance with western blotting. This work was supported by grants from the National Institute of General Medical Sciences (R01 GM089771 to JC and R35 GM122569 to TH).

## Author contributions
Conceptualization and experimental design by N.S., A.R.-O., B.J.L., T.H. and J.C.; experimentation, data collection and analysis by N.S., A.R.-O., M.A.C. and J.F.M.; resources by B.J.L. and T.H.; writing by N.S., A.R.-O., and J.C.; editing by N.S., A.R.-O., T.H. and J.C.; funding acquisition by T.H. and J.C.

## Competing interests
The authors declare no competing interests.
