## [Peer Review File · Nature Communications]

REVIEWER COMMENTS

Reviewer #1 (Remarks to the Author):

This is an innovative work addressing lipid-protein interactions that regulate a spectrum of cell function. Using a single-molecule cell-lysate pulldown assay and recursive-modeling, the authors have investigated the structural properties of PH-domain-containing proteins that allow their PIP binding. They have derived a predictive model based on 67 full-length PH-domains containing proteins and validated it with additional 22 proteins. The model predicts that nearly 50% of 222 human PH-domain containing proteins have PIP-binding ability, which is 5-times higher than previously thought. Taking ARHGEF3 as an example, the work has shown the unexpected importance of full-length protein in addition to its PH domain in PI(4,5)P2/P(13,5)P2 interactions. The experiments are well-designed with rigor. The work is expected to have a broad impact on cell signaling.

I have several questions below:

- 1) In the pull-down assay using PIP-containing liposome vesicles, whether and how does the nanoscale distribution of PIPs on these vesicles affect PH-domain interactions? To what extent such PIP spatial distribution mimics those in the native cell membrane?
- 2) It is well-known that both the domain structure and surface charge regulate the interactions between PH-domain containing proteins and lipid membrane. This work focuses more on the former, how does the surface charge impact the model prediction?
- 3) Discuss why full-length ARHGEF3 binds PIP2 but its stand-alone PH domain did not.
- 4) Vesicles containing 20 mol % PIPs pull out additional 10 proteins as compared to 5% PIPs, suggesting the importance of saturating PIP concentration in the assay sensitivity. This may be in particular critical for testing those proteins with low-affinity or co-incidence binding. However, most tests in this work using 5% PIP2, how does that underestimate the results and model prediction?
- 5) The single-molecule pulldown is quantified as binary results rather than gradient variables, however, the raw data (figure1) showed a clear correlation with the binding affinity. does the model with the binding affinity scores improve prediction accuracy?
- 6) Using full-length proteins from cell lysate is an advantage of the assay. How do those proteins containing multiple PH-domains bias the outcomes?

Reviewer #2 (Remarks to the Author):

Chen and coworkers determined the membrane binding properties of 67 full-length PH domain proteins by a vesicle pull-down assay called SiMPull, which this group reported in 2016. The concept of vesicle pulldown is not new but a new twist here is that binding to the vesicles is quantified by fluorescence single molecule counting using GFP-tagged proteins. Two main advantages of the method are improved sensitivity and an ability to use fully processed proteins while bypassing laborious protein purification. The authors made some interesting findings about the membrane binding of PH domain-containing proteins and developed a prediction algorithm for membrane binding PH domains. Overall, the work is interesting and can have broad impact in the field of lipid-protein interactions. However, there are some major issues with the current version of the manuscript that need to be addressed before it is seriously considered for publication.

Major issues:

1. Overall, the report is too descriptive and lacks rigor and structural and mechanistic insights. Perhaps the most important finding is that some proteins bind PIPs only as intact proteins. Although their PH domains are responsible for PIP binding, isolated PH domains (as GFP-fusion) do not bind PIPs. Unfortunately, the authors did not perform systematic structure-function analysis to understand the structural basis for the finding. It is well documented that PH domains have much lower sequence similarity than other lipid binding domains, such as FYVE domains. Thus, one reasonable possibility is that for some proteins the lipid binding site is composed of what is defined as the PH domain plus other parts of the proteins. Authors should select at least one protein and perform detailed studies (preparing multiple PH domain constructs with different N- and C-terminal extensions, etc) to fully address this issue.
2. Bypassing protein purification is obviously a major advantage but if this also serves as a source of low credibility, it has to be addressed. The cell extract contains a wide variety of materials that can affect the properties of proteins and some controls are necessary to address this point. For instance, is it related to the finding that for some proteins only FL proteins bind PIPs? At least for some proteins, they should demonstrate that membrane binding properties of purified FL proteins and PH domains are identical to those of unpurified proteins.
3. Since the authors developed a predictive algorithm, they should address the limitation of the binary information. Membrane binding of proteins is a complex process that involve multiple interactions and protein conformational changes, and protein-protein interaction. Also, binding equilibrium is governed by K_d value and analyte concentrations. I understand that for this type of

large-scale work it is not practical to control all these factors. However, to justify the criteria the authors used to classify binder and non-binder (especially for predictive algorithms), some real quantitative parameters, like K_d , must be provided and concentrations of total lipids and proteins must be defined to some degree. Otherwise, distinction between binder and non-binder can be arbitrary.

Minor issues:

1. PI(4,5)P₂ is more abundant than other phosphoinositides and it is mainly found in the PM. However, it is not highly abundant in the PM. Its concentration there was estimated about 1 mol%.

Reviewer #3 (Remarks to the Author):

The work by Singh et al. explores the ability and specificity of human PH domain-containing proteins to bind PIPs. Results presented here showcase that these properties of PH-domain containing proteins were highly underestimated to date. The authors have performed a plethora of experiments to support their hypotheses and have also developed a computational method that allowed them to investigate the PIP binding potential of all human proteins with PH domains. Results produced by their computational method were also supported with more experiments showcasing the method's high sensitivity to predict proteins that can bind PIPs, with a lot of potential future applications.

My primary area of research is not within biophysics, so in my comments below I have mainly focused into accessing the computational method developed in this work.

Major comment:

- The authors have performed experiments with 20 PH domain containing-proteins that their method predicted as PIP binding, which were never reported to bind any lipids experimentally before this work. In order to fully evaluate their method, the authors should make a similar test for a set of proteins predicted as not binding PIPs by their computational method, to make sure that proteins are not falsely classified as negatives, when they might also possess the ability to bind PIPs. For the time being they have showcased that the method developed has a good precision, but the recall is not proven yet. The experimental results presented by the authors in the preceding sections, clearly demonstrate that the ability of PH-domain containing proteins to bind PIPs was highly underestimated in the past (10% vs. 50% predicted by the authors). Thus, I believe that such

experiments for a set of proteins in the negative set is necessary to showcase that their computational method can be widely used to predict which PH containing proteins can bind PIPs with high accuracy and thus help guide more experiments to elucidate the nature of these proteins. Doing those experiments will also allow the authors to calculate metrics of their method's accuracy, which are currently not reported in the paper. If the method has a high accuracy it can also serve as a stepping stone for evolutionary studies on the nature of PIP binding of PH domain-containing proteins in eukaryotes, thus providing a very strong tool into understanding the nature of these interactions and maybe decipher why many proteins have lost/gained this ability over time.

Minor comments:

- add reference to Pfam database (e.g. pg. 3, line 6)
- there are many acronyms in the manuscript; I would suggest that acronyms that are not used a lot throughout the paper should be extended to their constituents, to make reading easier (e.g. LBDs for lipid binding domains used only twice).
- "We have now applied the lipid-SiMPull assay to interrogate tens of human full-length PH domain proteins for their binding to vesicles of various compositions", use "investigate the ability of ... to bind", instead of "interrogate ... for their binding"
- the authors refer to human PH domain proteins a lot of times in the text. Since, these proteins can have more domains than just PH, I would suggest to rephrase all instances to the more accurate PH domain-containing proteins throughout the text.
- "... predicts PIP binding for the entire family of human PH domains", maybe the "entire family of proteins with PH domains"?
- "Expression of each fusion protein in HEK293 cells was confirmed by western analysis", rephrase to "western blot analysis" or "western blotting".
- Table 1: I believe that it can help the readers if instead of sorting based on protein name, you instead sort the table from more specific (only 1 PIP bound from those tested) to least specific, to non-specific, and then add negatives with a clear distinction (e.g. a double line to distinguish them).
- Figure 1C: Instead of 1,2,3,4,5,6,7,8 under EGFP and DiD add the lipids, or add the numbers under the microscopy images for quick reference.
- Figure 3D: Show side chains in the structures for the 4 residues in magenta to see the interaction of those in 3D
- Supplementary Table 1: add some conditional formatting in the excel sheet with a color scale to indicate quickly positive and negative results (makes it easier to follow each line shown)
- Figure 5D: add sequence in the bottom of the figure, difficult to follow from 5C. Also, can you make any quick comments on the physico-chemical properties of the residues marked as important?

- explain the reason why you chose a method that also uses structural information to align PH domains (I believe the choice is valid, but should be briefly discussed in the manuscript)
- Supplementary Figure 4 is not showing for me in the online version of the manuscript
- I would like to suggest to the authors to make their computational method available through a software hosting service like github (allows for quick identification and better outreach)
- wrong beta in " β -glycerophosphate" in Methods section
- RFC Matrix Generation and PIP-binding Prediction (Methods): "about 30 sequences" are these 31? If yes write the number, "297 x 20" explain that 297 are the positions in the MSA and 20 are the amino acid residues.
- Supplementary Table 4: change font of aligned sequences to a monospaced font (e.g. Courier New)

We are excited by the reviewers' appreciation of the overall significance of our work. We have completed numerous new experiments based on the reviewers' constructive criticisms, and the manuscript has now been substantially revised. Below is a point-by-point response to the reviews. A copy of the manuscript text with significant modifications highlighted is also uploaded. We are very grateful for the reviewers' suggestions that have led to marked improvement of this manuscript.

A note: the total number of PH domains in the original manuscript was stated to be 246, but we have found 4 domains that were duplicated in the "not-assayed" list due to different names given to the same genes. Hence, the new total is 242. This did not change the RFC matrix or the outcome of the prediction.

Reviewer #1 (Remarks to the Author):

This is an innovative work addressing lipid-protein interactions that regulate a spectrum of cell function. Using a single-molecule cell-lysate pulldown assay and recursive-modeling, the authors have investigated the structural properties of PH-domain-containing proteins that allow their PIP binding. They have derived a predictive model based on 67 full-length PH-domains containing proteins and validated it with additional 22 proteins. The model predicts that nearly 50% of 222 human PH-domain containing proteins have PIP-binding ability, which is 5-times higher than previously thought. Taking ARHGEF3 as an example, the work has shown the unexpected importance of full-length protein in addition to its PH domain in PI(4,5)P2/PI(3,5)P2 interactions. The experiments are well-designed with rigor. The work is expected to have a broad impact on cell signaling.

I have several questions below:

1) In the pull-down assay using PIP-containing liposome vesicles, whether and how does the nanoscale distribution of PIPs on these vesicles affect PH-domain interactions? To what extent such PIP spatial distribution mimics those in the native cell membrane?

RESPONSE: Giant lipid vesicles have been reported to contain nanodomains that mimic native cell membranes, but we are not aware of any report of nanodomain characterization of the type of small unilamellar vesicles used in our assays. This is an important question that should be explored in future investigations. We have added this point to the discussion in the manuscript.

2) It is well-known that both the domain structure and surface charge regulate the interactions between PH-domain containing proteins and lipid membrane. This work focuses more on the former, how does the surface charge impact the model prediction?

RESPONSE: Our prediction algorithm simply associates any amino acid residues in the PH domain with PIP binding based on observations. The model does not specifically take surface charge into consideration, although one could assume that amino acids contributing to surface charge, which in turn impact PIP binding, would be identified by the algorithm. It would be interesting to see how surface charge distribution of the PH domains compare between the binders and non-binders and if a predictive model can be created using this information, but this is beyond the capability of our current modeling.

3) Discuss why full-length ARHGEF3 binds PIP2 but its stand-alone PH domain did not.

RESPONSE: Our previous ARHGEF3 PH domain construct was based on the sequence defined by Uniprot and included aa299-466. We have now remade the cDNA constructs using the crystal structure of ARHGEF3-PH as a guide (aa320-455), and found that the PH domain alone binds PI(4,5)P₂ but not PI(3,5)P₂. We have now performed additional domain analysis and gained better insights into the distinct structural requirements for ARHGEF3 binding to PI(4,5)P₂ and PI(3,5)P₂ (Figure 3C,D). Please see our response to Reviewer 2's point #1 for details.

4) Vesicles containing 20 mol % PIPs pull out additional 10 proteins as compared to 5% PIPs, suggesting the importance of saturating PIP concentration in the assay sensitivity. This may be in particular critical for testing those proteins with low-affinity or co-incidence binding. However, most tests in this work using 5% PIP₂, how does that underestimate the results and model prediction?

RESPONSE: The majority of studies in the literature use 5% PIP to represent lipid membrane in PIP binding assays. We would like to clarify that we did not find 10 more proteins that bind PIPs at 20%. We tested 10 proteins that had previously been reported to bind PIPs but did not bind in our assays with 5% PIP, and 4 of those 10 proteins bound 20% PIP. It is indeed likely that if we had screened all the proteins against 20% PIPs we might have discovered even more binders. However, even though some PIPs have been found to cluster in the cell membrane leading to high local concentrations, it is currently not clear under what physiological conditions and to what extent they occur. Therefore, we chose to use the conventional 5% so our overall data (50% of proteins bind PIP) can be viewed in the context of past findings (10% bind PIP).

5) The single-molecule pulldown is quantified as binary results rather than gradient variables, however, the raw data (figure1) showed a clear correlation with the binding affinity. does the model with the binding affinity scores improve prediction accuracy?

RESPONSE: Weighted contribution would not be trivial in a probability algorithm, and this becomes even more complicated when a protein binds more than one PIP with different affinities. We are not able to incorporate varying affinities into our model, which only considers whether a protein binds *any* PIP. Because of the variation in vesicle preparations containing different PIPs, we do not feel it is reliable to directly compare the number of GFP proteins pulled down across different PIPs.

6) Using full-length proteins from cell lysate is an advantage of the assay. How do those proteins containing multiple PH-domains bias the outcomes?

RESPONSE: If a protein contains two PH domains (or more) and does not bind any PIP, we put all the PH domains in the “no-binding” category (examples: AFAP1, FGD5, etc.). If such a protein binds PIP, we report the results in Table 1 but we do not include any of its PH domains for building the RFC matrix because we would not know which PH domain contributes to the binding (there are two such proteins on our list: ADAP1 and AFAPIL1). We have explicitly stated this in the revised manuscript.

Reviewer #2 (Remarks to the Author):

Chen and coworkers determined the membrane binding properties of 67 full-length PH domain proteins by a vesicle pull-down assay called SiMPull, which this group reported in 2016. The concept of vesicle pulldown is not new but a new twist here is that binding to the vesicles is

quantified by fluorescence single molecule counting using GFP-tagged proteins. Two main advantages of the method are improved sensitivity and an ability to use fully processed proteins while bypassing laborious protein purification. The authors made some interesting findings about the membrane binding of PH domain-containing proteins and developed a prediction algorithm for membrane binding PH domains. Overall, the work is interesting and can have broad impact in the field of lipid-protein interactions. However, there are some major issues with the current version of the manuscript that need to be addressed before it is seriously considered for publication.

Major issues:

1. Overall, the report is too descriptive and lacks rigor and structural and mechanistic insights. Perhaps the most important finding is that some proteins bind PIPs only as intact proteins. Although their PH domains are responsible for PIP binding, isolated PH domains (as GFP-fusion) do not bind PIPs. Unfortunately, the authors did not perform systematic structure-function analysis to understand the structural basis for the finding. It is well documented that PH domains have much lower sequence similarity than other lipid binding domains, such as FYVE domains. Thus, one reasonable possibility is that for some proteins the lipid binding site is composed of what is defined as the PH domain plus other parts of the proteins. Authors should select at least one protein and perform detailed studies (preparing multiple PH domain constructs with different N- and C-terminal extensions, etc) to fully address this issue.

RESPONSE: In the original manuscript we did perform domain analysis of ARHGEF3 in an attempt to understand its structural requirements for PIP binding, but we found its PH domain alone not to bind any PIP. Following the reviewer's suggestion, we re-examined the PH domain sequence and found a difference between the Uniprot definition we used to make our constructs and the reported crystal structure. We have now made new ARHGEF3 PH domain constructs using the structure as a guide (aa320-455). We have fused EGFP at either end of the PH domain in consideration of potential effects of the tagging. Our assay results show that the PH domain alone binds PI(4,5)P₂ but not PI(3,5)P₂ regardless of the tagging mode. However, when EGFP is tagged at the N-terminus, the PH domain becomes promiscuous and binds several additional PIPs (Figure 3C). We have performed further domain analysis to dissect the distinct structural requirements for ARHGEF3 binding to PI(4,5)P₂ and PI(3,5)P₂ (Figure 3D). The overall conclusion is that isolated PH domain does not have the same selectivity for PIP as the full-length protein. Specifically, (a) the PH domain is necessary and sufficient for ARHGEF3 binding to PI(4,5)P₂; (b) PI(3,5)P₂ binding requires the N-terminal domain in addition to the PH domain; (c) the C-terminus appears to suppress ARHGEF3 binding to both lipids; and (d) this suppression may be relieved by the N-terminal domain. We appreciate the reviewer's suggestion that has led us to gain deeper insights.

2. Bypassing protein purification is obviously a major advantage but if this also serves as a source of low credibility, it has to be addressed. The cell extract contains a wide variety of materials that can affect the properties of proteins and some controls are necessary to address this point. For instance, is it related to the finding that for some proteins only FL proteins bind PIPs? At least for some proteins, they should demonstrate that membrane binding properties of purified FL proteins and PH domains are identical to those of unpurified proteins.

RESPONSE: The EGFP-fusion proteins in our lysates were overexpressed, making it less likely that endogenous factors would reach stoichiometric levels to directly participate in the observed lipid binding. Nonetheless, the reviewer makes a valid point that direct lipid-protein interaction is not proven by our assays. We have made serious attempts to purify 11 of the full-length proteins found to bind PIPs in our hands, but even with the SUMO tag (known to help stabilize and solubilize recombinant proteins) and low temperature conditions for protein expression, the purified proteins did not stay soluble after purification. We then utilized bacterial lysates expressing the recombinant proteins directly in SiMPull assays. While 7 out of the 11 proteins were found to bind lipid vesicles nonspecifically, suggesting protein misfolding, 4 of the proteins were amenable to assay against PIP vesicles. Three of them recapitulated the PIP binding of their mammalian counterparts whereas the fourth, ARHGEF3, bound PI(4,5)P₂ but not PI(3,5)P₂. Taken together with the results of domain analysis described above, this result suggests that the full-length ARHGEF3 may require another factor to relieve an autoinhibition on PI(3,5)P₂ binding. It must be emphasized that an alternative and equally likely explanation would involve a posttranslational modification of ARHGEF3 necessary for PI(3,5)P₂ binding and not PI(4,5)P₂ binding. We would like to emphasize that the effectiveness of point mutations in the PH domain in abolishing PIP binding provides strong evidence for a direct interaction between ARHGEF3 and the specific PIPs.

3. Since the authors developed a predictive algorithm, they should address the limitation of the binary information. Membrane binding of proteins is a complex process that involve multiple interactions and protein conformational changes, and protein-protein interaction. Also, binding equilibrium is governed by K_d value and analyte concentrations. I understand that for this type of large-scale work it is not practical to control all these factors. However, to justify the criteria the authors used to classify binder and non-binder (especially for predictive algorithms), some real quantitative parameters, like K_d, must be provided and concentrations of total lipids and proteins must be defined to some degree. Otherwise, distinction between binder and non-binder can be arbitrary.

RESPONSE: The GFP-fusion proteins in cell lysates were always assayed at the same concentration (5 nM). Under such conditions we determined the threshold of detection to be K_d of 10-20 μM (Supplementary Figure 2). Non-binding is therefore defined as having a K_d larger than 20 μM.

Minor issues:

1. PI(4,5)P₂ is more abundant than other phosphoinositides and it is mainly found in the PM. However, it is not highly abundant in the PM. Its concentration there was estimated about 1 mol%.

RESPONSE: Thank you for pointing this out. We have now changed the wording from “highly abundant in the PM” to “mostly found in the PM”.

Reviewer #3 (Remarks to the Author):

The work by Singh et al. explores the ability and specificity of human PH domain-containing proteins to bind PIPs. Results presented here showcase that these properties of PH-domain containing proteins were highly underestimated to date. The authors have performed a plethora of experiments to support their hypotheses and have also developed a computational method that

allowed them to investigate the PIP binding potential of all human proteins with PH domains. Results produced by their computational method were also supported with more experiments showcasing the method's high sensitivity to predict proteins that can bind PIPs, with a lot of potential future applications.

My primary area of research is not within biophysics, so in my comments below I have mainly focused into accessing the computational method developed in this work.

Major comment:

- The authors have performed experiments with 20 PH domain containing-proteins that their method predicted as PIP binding, which were never reported to bind any lipids experimentally before this work. In order to fully evaluate their method, the authors should make a similar test for a set of proteins predicted as not binding PIPs by their computational method, to make sure that proteins are not falsely classified as negatives, when they might also possess the ability to bind PIPs. For the time being they have showcased that the method developed has a good precision, but the recall is not proven yet. The experimental results presented by the authors in the preceding sections, clearly demonstrate that the ability of PH-domain containing proteins to bind PIPs was highly underestimated in the past (10% vs. 50% predicted by the authors). Thus, I believe that such experiments for a set of proteins in the negative set is necessary to showcase that their computational method can be widely used to predict which PH containing proteins can bind PIPs with high accuracy and thus help guide more experiments to elucidate the nature of these proteins. Doing those experiments will also allow the authors to calculate metrics of their method's accuracy, which are currently not reported in the paper. If the method has a high accuracy it can also serve as a stepping stone for evolutionary studies on the nature of PIP binding of PH domain-containing proteins in eukaryotes, thus providing a very strong tool into understanding the nature of these interactions and maybe decipher why many proteins have lost/gained this ability over time.

RESPONSE: We have now assayed 11 proteins from the predicted non-binder list against all PIPs. We had intended to validate a larger number, but due to negative impacts of the pandemic our source of the cDNAs (DNASU) was not able to deliver some of them. Nevertheless, we have now performed validation on a total of 31 proteins, a reasonable number considering that our original data input to build the model came from 67 proteins. These results (Figure 7B,C) have led to the construction of a confusion matrix (Figure 7D) that yielded the following metrics: Accuracy = 0.87, Recall = 0.864, Precision = 0.95, F-1 score = 0.90. We are grateful to the reviewer for suggesting these experiments that have significantly strengthened our work.

Minor comments:

- add reference to Pfam database (e.g. pg. 3, line 6)

RESPONSE: Done.

- there are many acronyms in the manuscript; I would suggest that acronyms that are not used a lot throughout the paper should be extended to their constituents, to make reading easier (e.g. LBDs for lipid binding domains used only twice).

RESPONSE: We have gone through the text and made sure that an abbreviation is used only when it appears at least 3 times in the text.

- "We have now applied the lipid-SiMPull assay to interrogate tens of human full-length PH

domain proteins for their binding to vesicles of various compositions", use "investigate the ability of ... to bind", instead of "interrogate ... for their binding"

RESPONSE: We have modified the text as suggested.

- the authors refer to human PH domain proteins a lot of times in the text. Since, these proteins can have more domains than just PH, I would suggest to rephrase all instances to the more accurate PH domain-containing proteins throughout the text.

RESPONSE: We have modified the text as suggested.

- "... predicts PIP binding for the entire family of human PH domains", maybe the "entire family of proteins with PH domains"?

RESPONSE: We have modified the text as suggested.

- "Expression of each fusion protein in HEK293 cells was confirmed by western analysis", rephrase to "western blot analysis" or "western blotting".

RESPONSE: We have modified the text as suggested.

-Table 1: I believe that it can help the readers if instead of sorting based on protein name, you instead sort the table from more specific (only 1 PIP bound from those tested) to least specific, to non-specific, and then add negatives with a clear distinction (e.g. a double line to distinguish them).

RESPONSE: We have reorganized Table 1 following the suggestion.

- Figure 1C: Instead of 1,2,3,4,5,6,7,8 under EGFP and DiD add the lipids, or add the numbers under the microscopy images for quick reference.

RESPONSE: This figure is already over-sized and we are not able to find an efficient and effective way to add the labels as suggested. Since the order of PIPs represented by 1,2,3 etc in 1C is identical to the order for the microscopy images in 1B right next to 1C, and this is also explained in the figure legend, we hope it won't be very confusing for the readers.

- Figure 3D: Show side chains in the structures for the 4 residues in magenta to see the interaction of those in 3D

RESPONSE: This is now Figure 4A. We have added an additional ribbon structure showing the side chains while retaining the original ribbon structure for better visibility of positioning of the amino acids.

-Supplementary Table 1: add some conditional formatting in the excel sheet with a color scale to indicate quickly positive and negative results (makes it easier to follow each line shown)

RESPONSE: We modified the table as suggested.

-Figure 5D: add sequence in the bottom of the figure, difficult to follow from 5C. Also, can you make any quick comments on the physico-chemical properties of the residues marked as important?

RESPONSE: We have added the sequence as suggested (now Figure 6D). However, we must emphasize that this sequence is only a single example (AKT-PH) and it does not reflect the composite information derived from considering the behaviors of 67 PH domain as shown in the heat map in Figure 6C. For the same reason, the bars marked in the graph of Figure 6D do not represent particular residues but instead they indicate the important *positions* in the PH domain structure (e.g. β_3 , β_4 , α -helix) where certain amino acids significantly contribute to PIP binding or no-binding.

- explain the reason why you chose a method that also uses structural information to align PH domains (I believe the choice is valid, but should be briefly discussed in the manuscript)

RESPONSE: We have now added this in the manuscript: "PH domains are known to have low sequence conservation but they all share a conserved structure. The incorporation of structural

information is therefore particularly important for the accurate alignment of PH domain sequences.”

- Supplementary Figure 4 is not showing for me in the online version of the manuscript

RESPONSE: This figure was a very large file which was difficult to open sometimes. We have now reduced the size and fixed that problem.

- I would like to suggest to the authors to make their computational method available through a software hosting service like github (allows for quick identification and better outreach)

RESPONSE: We have now deposited the computational method at Github as described in Methods.

- wrong beta in "β-glycerophosphate" in Methods section

RESPONSE: Corrected.

- RFC Matrix Generation and PIP-binding Prediction (Methods): "about 30 sequences" are these 31? If yes write the number, "297 x 20" explain that 297 are the positions in the MSA and 20 are the amino acid residues.

RESPONSE: We now indicate the exact number (34) and explain the matrix as suggested.

- Supplementary Table 4: change font of aligned sequences to a monospaced font (e.g. Courier New)

RESPONSE: We have modified both Supplementary Tables 3 & 4 as suggested.

We greatly appreciate the reviewer's keen eye and excellent suggestions to improve the presentation of our work. Most of the wording changes described above were not highlighted in the marked copy of manuscript text.

REVIEWERS' COMMENTS

Reviewer #1 (Remarks to the Author):

This revised manuscript has addressed my points. Before acceptance for publication, the authors need to go through the figures to clarify the statistics (sample numbers/replicates or comparison tests used).

Reviewer #2 (Remarks to the Author):

The authors satisfactorily addressed all my concerns.

Reviewer #3 (Remarks to the Author):

I would like to thank the authors for satisfactorily addressing all the issues I have raised in my initial report. The limitation they mentioned in the additional experiments they carried out is acceptable, considering the ongoing global pandemic. I believe the additional experiments showcase the validity of the predictions of their computational method. I would also like to thank them for taking into consideration all the minor issues I have raised. I believe the authors have done an overall good job towards addressing comments raised by all the reviewers and the changes applied have improved their work.